# Method of Mathematical Modeling for the Experimental Evaluation of Flame-Retardant Materials’ Parameters

**DOI:** 10.3390/ma15010011

**Published:** 2021-12-21

**Authors:** Dmitry Alexandrovich Korolchenko, Tatiana Yurievna Eremina

**Affiliations:** Department of Integrated Safety in Civil Engineering, Moscow State University of Civil Engineering, Yaroslavskoye Shosse, 129337 Moscow, Russia; a.barvina@ikbs-mgsu.ru

**Keywords:** fire retardancy, fire protection, experimental research, experimental design, mathematical methods, input and output parameters

## Abstract

Mathematical and experimental research plays an active role in fire protection investigation. The choice of optimal conditions for the experimental program is the main methodological part of this research. The peculiarities of new fire-protective compositions were investigated. Many experiments in this work are aimed at the investigation of the physical and chemical properties of the materials.

## 1. Introduction

The variety of modern methods enables optimal planning of experiments. The investigation process is divided into several phases. In each phase, the researcher receives new data that enable them to adapt the program of experimental research. In accordance with the experimental research, the essence of the tests is in the choice of input data for the experiment. In each phase of the research, the selection of the optimal position of the points in the factor space should be made in order to obtain an optimal area for mathematical modeling. After defining the optimal zone, the task is formulated differently. Here, the investigator must understand the response surface much better, by approximating it with polynomials of the second and sometimes even third degree. Frequently the experimenter faces screening experiments, which aim to emphasize the influence of a multitude of possible effects [1,2,3,4,5].

Recently, due to the increase in construction volumes, as well as the construction of unique structures, the requirements for fire resistance of building structures have increased significantly. In this regard, the construction industry has shown interest in new fire-retardant coatings, as well as in steels capable of maintaining sufficient safety margins during short-term heating in fire conditions while simultaneously meeting all of the operational requirements applicable to metal structures [6,7,8,9].

The fire resistance of metal structures in fire conditions depends on many factors, principal among which are the stress–strain state, the fire intensity, and the structure’s methods of fire protection. Various methods of fire protection are used to increase the fire resistance of building structures, such as concrete coating, fire-retardant cladding, fire-retardant coatings, etc. The load-bearing frame of high-rise buildings is usually designed with monolithic reinforced concrete and/or steel structures, with fire protection of their structural materials, and the durability of fire protection should correspond to the design life of the building until total renovation [10,11,12,13,14,15].

The performance properties of fire-retardant coatings are directly related to the service life of the buildings, as treated steel structures are often covered with plasterboard or other materials after the fire-retardant coating is applied, and access to these coatings is permanently closed, or else rather complicated removal of wall panels is required.

Moreover, due to the fact that the service life of buildings and structures is measured in decades, there is also the issue of preserving the fire protection effect of coatings in the course of long-term operation. The importance of addressing this issue becomes evident when one assumes that the fire protection effect may be partially or completely lost over time, without any visible change in the coating itself.

Passive protection measures are implemented to ensure the required fire resistance rating of structures, one of them being the application of special coatings on the surface of structures, the purpose of which is to form a low-thermal-conductivity screen when heated, protecting the metal from heating and destruction [16,17].

Therefore, it is necessary to investigate the properties of flame retardants, in order to reduce their cost and improve the properties that make it possible to apply them mechanically—especially on complex structures and in hard-to-reach places—while also ensuring that they meet aesthetic requirements and do not release toxic components.

Currently, a wide range of intumescent flame retardants is used both in Russia and abroad to increase the fire resistance rating of steel structures.

A number of Russian intumescent coating studies—which raised not only the issue of creating flame retardants, but also the behavioral model of flame retardants under the effects of fire—are worth noting. Russian studies also focus on such issues as the prediction of the service life of flame-retardant coatings [18,19,20].

The majority of foreign studies refer to compositions that do not emit toxic gases when heated, mainly concerning compositions containing either water or inorganic fillers (e.g., vermiculite, asbestos). It should be noted that the new intumescent coatings produced on the basis of aqueous dispersions are characterized by low toxicity and low intensity of odor emissions. German researchers noted that materials containing melamine have low toxicity [21,22,23,24].

The main source of information on flame retardant studies is patent specifications. As shown by patent searches, a significant number of intumescent flame retardants based on polyvinyl acetate, acrylic styrene, and other coupling agents have been developed. Such intumescent flame retardants are applied with a thickness of approximately 1 mm in order to achieve a fire resistance rating of 45 min. The intumescence of such coatings occurs in the temperature range of 200–500 °C, forming a porous thermal insulating layer with an intumescence ratio of 40–60 times. The resulting thermal insulation layer has low thermal conductivity due to the formation of a cellular structure filled with low-thermal-conductivity gases [25,26,27].

Japanese researchers offer a waterborne paint with viscosity of 50–1000 Pa·s, which contains a 100 ppm mixture (pH 7.5–9.5) consisting of (1) 2–15% (in terms of dry residue) acrylate- or methacrylate-based copolymer (CPL) emulsion containing the C1-8 alkyl group, or CPL of said (meth)acrylates and <30% styrene; (2) 1–5% (in terms of dry residue) dian epoxy emulsion; (3) 25–70% filler; (4) 20–60% α- or β-hemihydrate plaster; (5) 1–5% color pigment; and (6) 0.8–1.2 eq. polyamine or alicyclic polyfunctional amine as a component hardener (2) and, if necessary, small amounts of other additives, the components being in the weight ratio of [(1) + (2)]/(4) = 0.15–0.28 and [(2) + (6)]/(1) + (2) + (5) = 0.2–0.45. The minimum component film-formation point (1) is <5 °C. The paint is used for spray-coating of construction materials. The coating provides high fire protection efficiency and durability.

The proposed flame-retardant paints and mastics are composed of the following: disperse binders, e.g., polymers and copolymers of alpha- and beta-ethylated monomers and resins (5–50% wt.); substances charring when heated, e.g., aliphatic and alicyclic polyatomic alcohols, polysaccharides, proteins, aminoaldehyde condensates, haloid hydrocarbons, and their derivatives (4–50%); substances emitting nonflammable gases, e.g., carbonic acid amines, cyanamide and its oligomers, and isoamides of carboxylic acids (6–50%); charring catalysts, e.g., boric and/or phosphoric acid and their chemical compounds (3–40%); oxygen-containing molybdenum compounds, e.g., ammonium phosphomolybdate (0.03–5.0%), combustion inhibitors e.g., antimony and bismuth compounds (up to 12%); and, finally, pigments, additives, and fillers to obtain the desired color and consistency.

Nullifire Ltd. (London, UK) has developed a number of new flame retardants for building structure applications. In particular, the System-S607 material is low in toxicity, characterized by low intensity of odor emission, and is safe for handling; therefore, it can be freely used in food and other industrial plants, in apartment buildings, and in public and commercial multipurpose buildings.

Flakolight Paints (Tamil Nadu, India) is offering a new series of intumescent flame-retardant paints. These paints are waterborne, non-toxic, and are capable of producing class 1 and 0 surface flame spread materials in accordance with the British Standard BS 476 parts 6 and 7 tests. Unlike most manufactured intumescent flame-retardant paints, the new series includes a special additive that extends their shelf life to two years [28,29,30]. The paints are available in different colors, including black, white, or transparent lacquer finishes, and can be applied using a brush or paint sprayer [31,32].

Russian authors have developed a mathematical model, algorithm, and program for thermal engineering calculation of water-containing fire retardants, allowing in particular for the recording of the effect of continuously protected surface temperature stabilization with steady increase of ambient temperature (e.g., when tested according to the standard temperature conditions) during numerous fire tests. A distinctive feature of the model is the direct consideration of the kinetics of material dehydration and moisture condensation in the intrapore space, as well as the effect of variable moisture content of the fire retardant on its thermophysical properties. This enables the developed algorithm and program to be used in the optimization of fire protection of various structures, including under actual fire conditions. The results of numerical calculations sufficiently correlate with the experimental data of both tests of simple structure fragments with fire protection on the radiant heating bench and tests of full-scale building structures in the Federal State Budgetary Establishment All-Russian Research Institute for Fire Protection of the Ministry of the Russian Federation for Civil Defence, Emergencies and Elimination of Consequences of Natural Disasters (FGBU VNIIPO EMERCOM of Russia) fired heater (FGBU VNIIPO EMERCOM of Russia, Moscow, Russia). This confirms the possibility of practical application of the developed methodology of thermotechnical calculations of water-containing fire retardants [33,34,35,36,37,38].

The properties of the intumescent flame retardants on the market differ in many aspects, such as intumescence rate, activation temperature, the moisture content of the composition in its initial state, and a number of other parameters. The nomenclature of existing intumescent flame retardants is fairly wide, and has a tendency to increase further [39,40].

In accordance with GOST 12.1.044-89, polymers are materials capable of igniting when exposed to an ignition source and burning up independently after the source is removed. Polymer combustion is a complex physical and chemical process involving chemical reactions of degradation, crosslinking, and carbonization of the polymer in the condensed phase, chemical reactions of conversion and oxidation of gaseous products, and physical processes of intense heat and mass transfer. A large number of chemically diverse polymers are used for the production of intumescent flame retardants; therefore, it is relevant to study the behavior of polymers under the effect of the ignition source, and to determine the quality and quantity of the emitted toxic gases.

The creation of new fire protection agents requires solving a number of complex physicochemical issues of binders and fillers at standard ambient temperatures and humidity, and at high temperatures under fire conditions, as well as thermodynamics, solid-phase reactions, heat and mass transfer in capillary-porous bodies, and the mechanics of rigid bodies.

In addition, flame retardants must not only provide the required fire protection for the protected structure under external flame impingement, but also ensure good adhesion to the substrate material or structure, the required durability under normal operating conditions, manufacturability, and application to the protected structure [41,42,43].

Thus, in order to obtain a certain set of properties of the flame retardant, the following important parameters affecting the fire protection and performance properties must be considered:

The ratio between the binder and the pigment content largely determines the physicomechanical and protective properties of the coating. This value is usually referred to as the pigment volume concentration (PVC) value. At PVC ~45%, all properties undergo the most significant change—vapor permeability increases sharply and film strength decreases. This is commonly referred to as the critical PVC value, and is designated as CPVC. The CPVC value of aqueous compounds depends on the nature of the binder and the ability of latex particles to coalesce, along with the wetting properties of the pigment surface by the polymer and the shape and size of the pigment particles. If the maximum amount of pigment and/or fillers needs to be introduced, and a good latex film coating with the usual set of strength parameters (which are more frequently required in coating technology) is needed, a hydrophilic pigment and/or fillers, an adsorption-saturated binder, and a well-stabilized aqueous dispersion of pigment and fillers (paste) should be used.

The rheological properties of solutions (e.g., viscosity, plasticity) are primarily connected with the structure and chemical nature of the material [44,45,46,47,48]. The viscosity of coating materials is determined by internal friction between their layers when moving under the influence of external forces. The need to determine the viscosity index is due to the fact that coating materials with poorly selected viscosity are difficult to apply, and often result in coating surface defects (such as dimples and skips), leading to the loss of strength (performance) and flame-retardant properties of the coating. The application method also depends on the viscosity value, e.g., highly crosslinked coating materials are not suitable for application by dipping and pouring, because the excess paint does not drain away from the surface; they can be successfully applied by methods that provide high tensions or shear rates, such as spraying, brushing, and especially roller-coating methods. Most flame retardants are highly crosslinked due to the high content needed to impart flame-retardant properties [49].

Thus, extensive research is required in order to create a product with stable flame-retardant and performance properties when creating aqueous flame retardants.

## 2. Experimental Section

A study was carried out on changes in the effectiveness of aqueous-based flame retardants for steel structures with different component content percentages [50,51,52,53].

Our experimental research was aimed at the evaluation of the reliability of fire-retardant compounds under heat exposure, depending on the percentages of components in the formulation.

We considered changes in the fire-protective efficiency of coatings for steel constructions with various percentages of components in the formulations of their fire-retardant coatings.

Our results demonstrate that fire retardants for steel constructions increase their fire resistance limit up to approximately 60 min, compared to 15 min for unprotected constructions at steel’s critical temperature of 500–550 °C. Moreover, fire-protective efficiency significantly depends on the formulation [54,55,56,57]. The best fire-protective efficiency was demonstrated by the composition of 55% liquid glass, 25% serpentinite, 5% mica, and 15% polystirol.

When increasing the liquid glass percentage up to 70% and mica up to 11%, while decreasing polystirol down to 1%, the fire resistance of the composition was equal to only 34 min, i.e., almost half that of the composition with the formulation mentioned above.

Further analysis of fire-protective efficiency demonstrated its dependence on the percentages of liquid glass (optimal percentage of 60–65%), polystirol (9–11%), and serpentinite (12–15%); these percentages also influence the intumescence ratio.

Studies of small structure and material models were carried out in order to determine the reliability of the developed flame retardants under heat exposure, depending on the different percentage ratios of the components included in the flame-retardant formulation.

Our results show that the developed flame retardant enables an increase in the fire resistance rating of steel structures up to 60 min, compared to 15 min for unprotected structures, at the critical temperature of steel (500–550 °C) (Figure 1). Earlier it was determined that the composition with the optimal mixing ratio of the components responsible for intumescence had the best flame retardant efficiency. In addition, this composition contains the optimal amount of titanium dioxide for the optimal intumescence ratio.

Optimality means that the intumescent foam does not slip off under its weight during fire exposure, and the necessary flame retardant performance is ensured at the same time (Figure 2).

The figure shows that flame retardant 3 possesses the best flame retardant efficiency. The best component content percentage is the following: phosphorus compound: intumescent agent: carbonizing agent = 2:1:1. Results of intumescence ratio studies of coatings with different percentages of aqueous flame-retardant components are presented in Table 1 and Table 2.

The main component of any flame retardant is the binder. Binders are natural and synthetic resins and other high-molecular-weight compounds capable of forming a continuous film with residual hardness, strength, and elasticity, adhesion to the substrate and the top layers of the coating, resistance to moisture, etc., on a solid substrate under certain conditions. Industrial synthetic latexes have a particle size of up to 0.25 µm. At 50% dry substance content, this corresponds to 1 mL of such latex having 1013 particles, with a specific surface area of 7.5 m^2^/mL (spherical particles).

The selection of a binder for a particular coating is based primarily on its performance properties. The nature of the binder determines the production technology and the basic properties of the paint coating.

Flame retardants based on polyvinyl acetate dispersions are sufficiently lightfast and weatherproof.

Flame retardants containing polyvinyl acetate dispersions can be applied over GF-021 GOST 25129-82 primer (International Scientific Innovative Center of Construction and Fire Safety, Saint-Petersburg, Russia) on almost any type of surface (metal, wood, concrete, etc.). The main properties of the most commonly used polyvinyl acetate dispersions of various grades are shown in Table 3.

Aqueous flame retardants are multicomponent and, therefore, more complex and less stable systems than the synthetic latexes used to produce them.

Aqueous flame retardants contain a number of surfactant components acting as stabilizers to prevent binder particles and pigments from sticking or sedimenting. However, immediately after the manufacture of aqueous flame retardants, processes begin to take place due to the loss of kinetic and aggregative stability of the disperse phase components, including partial coagulation of latex particles, flocculation of pigment particles and sedimentation, and biological deterioration of some organic components, leading to changes in technological characteristics (e.g., decrease in viscosity, the appearance of odor).

The goal of the formulation of an aqueous flame retardant is to select additives that provide the paint with sufficient stability and good technological properties, and that form a film with the most closed structure possible.

Dispersion film formation is understood as a process of adhesion of the dispersed-phase particles when the dispersion medium is removed—for example, as a result of evaporation, with the formation of a single-phase continuous film. Based on the consideration of the laws of film formation from aqueous polymer dispersions, it is possible to formulate provisions that must be considered when selecting binders for aqueous flame retardants and conditions of coating formation. The mobility of the polymer chain at the film formation temperature has a significant (if not the decisive) influence on the film-forming ability of aqueous dispersions. Generally, latexes of polymers with T_C_ and a minimum film-forming temperature below room temperature are used as binders for aqueous flame retardants.

Polyvinyl acetate dispersion is the most common and simple binder. This prevalence of polyvinyl acetate paints is attributed to the ease of producing aqueous polymer dispersions and the relative cheapness of the monomer, as well as the suitability of polyvinyl acetate flame retardants for both interior and exterior coatings. 

The factors determining the effectiveness of flame retardants depend significantly on the chemical structure of their constituent components and the nature of the binder.

A compound demonstrates optimal properties (the lowest mass loss during combustion, high phosphorus(V) oxide content of over 72%, and the lowest nitrogen content of at most 15%) among the intumescent components available on the market of phosphorus compounds.

The intumescence ratio and flame retardant efficiency were determined for the obtained flame retardant samples. The obtained intumescence ratio and temperature-induced mass loss results are given in Table 4.

Thus, the Table 4 shows that the compositions with the best intumescence ratio indicators include PC with the following indicators: mass fraction of nitrogen of no more than 15%, and mass fraction of phosphorus pentoxide of 69–72%. The extreme values of the aforementioned indicators were checked to confirm this hypothesis (Figure 3).

We propose a mathematical model for the optimization of fire retardant formulations (various quantitative combinations) and fire retardant properties.

The input parameters are quantitative combinations (different component content percentages) in the compound formulation, while the output parameters are the properties (e.g., intumescence, adhesion, fire retardant efficiency). Pigments influence the intumescence ratio.

The choice of pigment is based on its properties (Table 5), including contact angle, TGA, and hydrophilicity. The film adhesive (latex film) is chosen to provide proper coverage.

For example, a good latex film covering can be produced within a hydrophilic pigment that stabilizes water dispersion. The pigment, in turn, influences the intumescence ratio. Therefore, we analyzed intumescent systems. Determination of hydrophilicity was reduced to the determination of the contact angles for the main fillers used in intumescent fire retardants.

For the introduction of the maximum amount of pigment and the formation of a good latex film coating with the usual set of strength parameters (which is more frequently required in coating technology), a hydrophilic pigment, an adsorption-saturated latex, and a well-stabilized aqueous pigment dispersion (paste) should be used. Further research was aimed at determining the hydrophilicity of traditional intumescent system fillers, which are divided into four groups: (a) polyols—organic hydroxyl compounds with high carbon content; (b) inorganic acids or substances that release acid at 100–250 °C; (c) organic amines or amides; and (d) halogenated compounds. The determination of hydrophilicity was reduced to determining the contact angles of the main fillers used in intumescent flame retardants. The results are presented in Table 5.

The properties of the main components used in intumescent coatings are presented in Table 6, Table 7, Table 8 and Table 9.

The experiments showed that the phosphates, foaming agents, and polyols considered in this work are hydrophilic fillers. Titanium dioxide as a pigment is also hydrophilic.

Therefore, the CPVC can be increased up to 65%, but according to the reference data for polyvinyl acetate dispersions with polymeric particle size of 0.5–10 μm, the CPVC is 45%. All intumescent flame retardants are overfilled systems; their CPVC is approximately 60%, which is necessary to achieve a certain flame retardant efficiency. The developed flame retardant was tested for the whole range of physical and mechanical characteristics, and it was established that with a CPVC of 60% the strength characteristics remained high, but the water solubility increased, which only allows for the obtained flame retardant to be used indoors, or with an upper weatherproof coating.

In the process of flame retardant formulation, it was determined that the flame retardant efficiency also depends on the ratio of the components included in the intumescent system. Different ratios of components were considered: carbonizing agent: PC: foaming agent = 1:2:1; carbonizing agent: PC: foaming agent = 1:1:1; carbonizing agent: PC: foaming agent = 1:2:2; and carbonizing agent: PC: foaming agent = 2:2:1.

Further selection of components was carried out based on the stability of the resulting dispersions (Table 10).

However, despite the precautions taken in the selection of components affecting the stability of the dispersions, the overall pH of the intumescent flame retardant was within 5–6, which made it difficult to select a thickener. Polyelectrolyte thickeners work only in an alkaline environment of pH up to 8–10, so in our case it was necessary to select a thickener that would be independent of the environmental pH, i.e., related to non-ionogenic polymers.

As a result of the study, a mathematical model was selected to determine the basic performance properties of the flame retardant.

## 3. Results and Discussion

Evaluation and prediction of fire retardant behavior (practically used or developed), with definite service time expiration, requires a set of special characteristics.

Therefore, intumescence research and coating thickness measurement are carried out on the object in order to predict fire-protective efficiency. All of the parameters mentioned above are determined by sampling the coating exactly at the object (from the material layer), where fire protection was provided, and comparing the values with those estimated in model investigation (after the research using an artificial aging method).

Coating structure changes were investigated before and after tests using the artificial aging method, based on real processing data, by sampling the coating at the object in St. Petersburg, where fire protection was provided, in order to predict the fire-protective efficiency of the composition.

Experimental research was based on mathematical modeling.

To construct a mathematical model of a complex, multifactor, multiparameter system, the following actions are required:1.Influencing factors (IFs) [x_1_, …, x_m_] and output parameters (OPs) [y_1_, …, y_n_];2.Make a plan of active multifactorial tests in the form of matrix X, containing m columns (according to the number of IFs) and N rows (according to the number of tests), the main requirements for which are as follows:(a)Lack of correlation between IFs (pair correlation coefficient r_xkl_ between factors xk and xl should be close to 0);(b)The completeness of the factor space coverage (which should be at least: N > m);(c)Practicability, i.e., compliance with the capabilities of the experimental bases;(d)All experiments (combinations of IFs) in matrix X are equivalent.3.Run active tests during which IF combinations are varied according to the plan (matrix X) and determine (measure) the values of the IFs, thus forming matrix Y, containing N rows (according to the number of tests) and n columns (according to the number of OPs). In this case, the results must be unambiguous, i.e., when repeating an experiment (reproducing the same IF combination), the deviation of the OP values should be insignificant;4.Carry out mathematical processing of the active test results, which involves the following:(a)Determining the relationship between the OPs by calculating the OP pair correlation coefficient (r_ykl_ values between parameters y_k_ and y_l_ must be close to 0, otherwise one of the OPs at y_k_ or y_l_ can be replaced by another OP);(b)Assessment of the correspondence of the sample of experimental values of each j-th OP [y_j1_, …, y_jN_] to normal (Gaussian) distribution, especially in accordance with the asymmetry coefficients As and kurtosis Ex (i.e., condition As = Ex = 0);(c)Building an adequate mathematical model:
(1)$$\;y_{j}=f_{j}\left(x_{1},\ldots ,x_{m}\right)\;jЄ\left[1,n\right],$$
which in this paper will take the form of a quasilinear equation regression:(2)$$y_{j}\approx \sum_{k=1}^{\;M_{j}}\;a_{jk}\;Z_{jk},\;jЄ\left[1,n\right],$$
where $a_{jk}$ is the regression coefficient, which is a component of the vector A_j_,$Z_{jk}$ is the k-th conditional factor, which is a component of the matrix $Z_{j}$, and represents the IF function $x_{1},\ldots ,x_{m}$, and *M_j_* is the number of regression coefficients or conditional factors ($M_{j}$ < N);(d)Using regression Equation (1) for applied purposes:Interpretation of the dependence of OPs on IFs;Evaluating the values of OPs for combinations of IFs that differ from those included in matrix X;Assessment of the significance of the influence of IFs on OPs;Construction of the working area on IF sets in which each j-th OP is within acceptable limits.

Conditional factors ($Z_{jk}$) are selected using the *accelerated choice method* as part of constructing the regression Equation (2), and the vectors of the regression coefficients A_1_, …, A_m_ are calculated based on the condition of minimum variance for regression equations (least squares method):(3)$$D_{j}={\left(N-M_{j}\right)}^{-1}\sum_{i=1}^{\;N}{\left(y_{ji}^{e}-y_{ji}^{b}\right)}^{2}\to min,\;jЄ\left[1,n\right],$$
where $y_{ji}^{e}$ and $y_{ji}^{b}$ are the values of the j-th OP, obtained during the i-th experiment and calculated using the regression Equation (2) for the i-th IF combination.

The adequacy of the regression Equation (2) can be assessed using the Fisher criterion.

The use of the *multimodal principle*, according to which the dependence of the j-th OP on the IFs can be described not by using one adequate Equation (2), but by using several such equations, also seems expedient.

It was necessary to construct a mathematical model of a complex system in the form of quasilinear regression Equation (2), containing four OPs (n = 4) influenced by eight IFs (m = 8). Matrix X was constructed for this specialized design of nine tests (N = 9).

Pre-processing of matrices X and Y made it possible to determine the following pair correlation coefficients (Table 11):$$r_{x12}=r_{x13}=r_{x14}=r_{x15}=r_{x16}=r_{x17}=r_{x23}=r_{x24}=r_{x25}=r_{x26}=r_{x27}=r_{x34}=r_{x35}=r_{x36}=r_{x45}=r_{x46}=r_{x47}=r_{x56}=r_{x57}=r_{x67}=1,0$$
*r*_y12_ = −0.276; *r*_y13_ = −0.231; *r*_y14_ = 0.173; *r*_y23_ = 0.703; *r*_y24_ = −0.576; *r*_y34_ = −0.841.

y1: expansion ratio;

y2: adhesion;

y3: durability;

y4: water resistance.

Four regression equations valid for the 1st OP were obtained in accordance with the multimodal principle:(4)$$y_{1a}\approx 4.622z_{1a}+5.652z_{2a},$$
where z_1a_ = (**x_1_x_2_x_3_**)^2^**x_5_**/**x_6_**; z_2a_ = (**x_4_x_7_**)^2^;
(5)$$y_{1b}\approx 9.407{\;z}_{1b}+6.537{\;z}_{2b},$$
where z_1b_ = **x_4_**/(**x_5_x_6_**); z_1b_=**x_1_x_2_x_4_x_5_x_6_x_7_**;
$$y_{1c}\approx 12.97{\;z}_{1c}-0.6064{\;z}_{2c}+6.019z_{3c},$$
where z_1c_ = 1; z_2c_ = **x_3_**^2^/(**x_5_**^2^**x_6_x_7_**); z_3c_ = (**x_1_x_2_x_4_**)^2^;
(6)$$y_{1d}\approx -4.066{\;z}_{1d}+1.379{\;z}_{2d}+3.892{\;z}_{3d}+4.172{\;z}_{4d}+1.023{\;z}_{5d},$$
where z_1d_ = **x_2_**^2^**x_3_x_5_x_6_x_7_**/**x_1_**; z_2d_ = **x_1_x_6_x_7_**/(**x_2_x_3_**^2^**x_4_**); z_3d_ = **x_1_x_3_**(**x_2_x_4_x_6_**)^2^/**x_7_**; z_4d_ = **x_2_x_3_x_4_**^2^**x_7_**/(**x_1_**^2^**x_5_**^2^**x_6_**); z_5d_ = **x_1_x_2_**^2^**x_3_x_4_**^2^**x_5_**^2^**x_6_**.

The experimentally obtained values of the OPs for the 2nd system were compared with the results of calculations using the regression equations (Table 12).

Values of the fire retardant efficiency of the material were examined for different buildings.

The methods chosen provide formulation development and experimental research for the properties of the new aqueous intumescent fire retardant. The results of the research are quantitative values: fire retardant efficiency—45, 60, or 90 min; intumescence ratio—20–40; heat conductivity—0.991 Wt/mK; film rinsability—3.5 g/m^2^.

A mathematical multifactorial model was developed, and the quantitative estimation of the influence of the components and their combinations on the main operating characteristics of the aqueous intumescent fire retardant was achieved.

A complex of formulations for fire retardants with stable properties was developed based on the results of the investigation of their components.

The operating properties of the materials were determined to provide fire protection.

## Figures and Tables

**Figure 1 materials-15-00011-f001:**
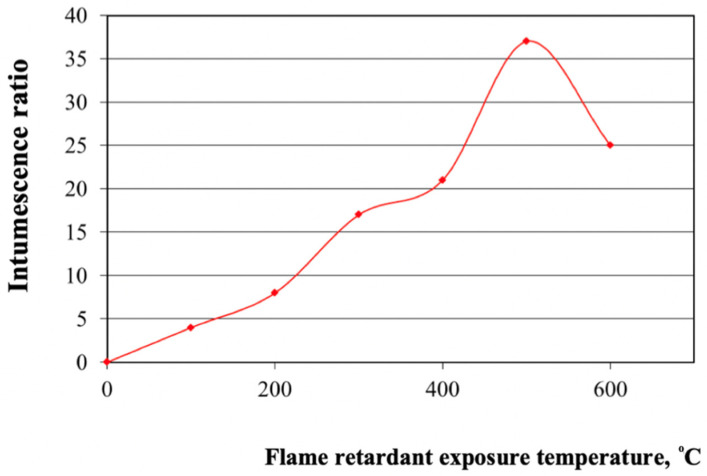
Dependence of the intumescence ratio on the flame retardant exposure temperature.

**Figure 2 materials-15-00011-f002:**
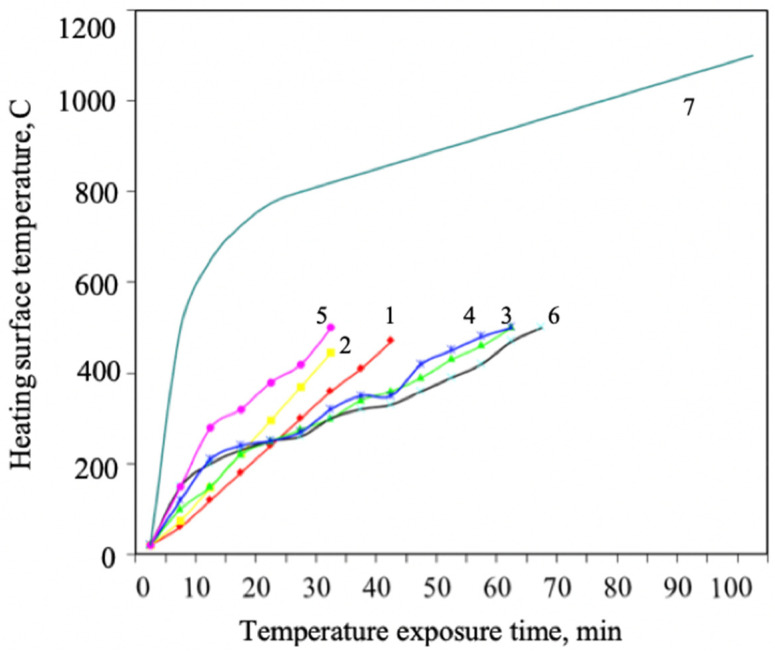
Changes in flame retardant efficiency depending on temperature for formulations with different component content percentages: 1—composition 1; 2—composition 2; 3—composition 3; 4—composition 4; 5—composition 5; 6—composition 6; 7—standard heater temperature conditions.

**Figure 3 materials-15-00011-f003:**
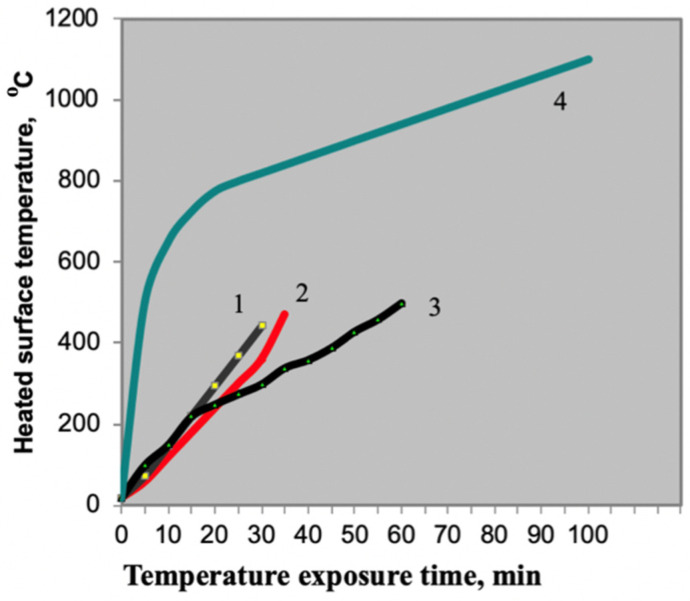
Flame retardant performance with the use of different phosphorus compounds: 1—PC Sample JLS-APP 102 formulation (China); 2—PC ANTIBLAZE formulation (France); 3—PC Exolit AP 422 formulation (Germany); 4—standard heater temperature conditions.

**Table 1 materials-15-00011-t001:** Retardants and their composition (comp.).

Component	Percentage					
	Comp. 1	Comp. 2	Comp. 3	Comp. 4	Comp. 5	Comp. 6
Water	19.7	23.8	20.1	20.1	16.5	20.1
Thickener	0.4	0.4	0.4	0.2	0.2	0.2
Dispersant	0.2	0.2	0.3	0.3	0.3	0.3
Surfactants	0.8	0.6	0.8	0.8	0.7	0.8
Defoamer	0.4	0.4	0.4	0.4	0.4	0.4
Coalescing agent	1.5	1.5	1.9	1.9	1.9	1.9
Titanium dioxide	9.0	6.9	8.3	8.3	8.3	8.3
Carbonizing agent	6.0	9.6	12.0	12.0	11.1	12.0
Intumescent agent	7.0	5.4	12.0	12.0	12.9	12.0
Phosphorus compound	28.0	21.4	24.0	24.0	22.9	24.0
Polyvinyl acetate dispersion	20.0	20.0	20.0	20.0	22.4	-
Chloride-containing polymer	7.0	5.4	-	-	-	-
Acrylic dispersion	-	-	-	-	-	20.0

**Table 2 materials-15-00011-t002:** Flame retardant thermogravimetric analysis results.

Retardant	Thermal Effects
Comp. 1	Endothermic effect in the temperature range of 25–300 °C. Extreme point temperature = 223 °C. The maximum reaction rate temperatures are 81 °C (water removal), 126 °C (pentaerythritol decomposition), and 223 °C. Exothermic effect in the temperature range of 300–418 °C. Extreme point temperature = 339 °C. Maximum reaction rate temperature = 383 °C. Exothermic effect in the temperature range of 418–741 °C. Extreme point temperature = 694 °C. The maximum reaction rate temperatures are 456, 552, and 594 °C (full phosphorus compound decomposition). Mass loss at 418 °C Δm = 48.9%. Mass loss at 601 °C Δm = 82.7%. Total mass loss at 741 °C Δm_total_ = 98.6%.
Comp. 2	Endothermic effect in the temperature range of 26–291 °C. Extreme point temperature = 184 °C. The maximum reaction rate temperatures are 81 (water removal), 140, 188 (pentaerythritol decomposition), 234, and 280 °C (beginning of phosphorus compound decomposition). Endothermic effect in the temperature range of 291–398 °C. Extreme point temperature = 328 °C. The maximum decomposition rate temperatures are 328 and 387 °C (melamine decomposition). Exothermic effect in the temperature range of 398–672 °C. Extreme point temperatures = 450, 532, 577 °C. The maximum reaction rate temperatures are 566, 601, and 642 °C (full phosphorus compound decomposition). Mass loss at 398 °C Δm = 65.4%. Mass loss at 601 °C Δm = 84.9%. Total mass loss at 673 °C—98.2%.
Comp. 3	Endothermic effect in the temperature range of 30–277 °C. Extreme point temperatures = 130 and 178 °C. The maximum reaction rate temperatures are 117 (water removal), 191, 215, 239, and 277 °C (pentaerythritol decomposition and beginning of phosphorus compound decomposition). Endothermic effect in the temperature range of 277–380 °C. Extreme point temperature = 338 °C. Maximum reaction rate temperatures = 326, 351, and 363 °C (melamine decomposition). Exothermic effect in the temperature range of 380–683 °C. Extreme point temperature = 548 °C. The maximum reaction rate temperatures are 474, 511, 524, 573, and 671 °C (full phosphorus compound decomposition). Mass loss at 380 °C Δm = 75.0%. Mass loss at 573 °C Δm = 90.0%. Total mass loss at 683 °C Δm_total_ = 96.5%.
Comp. 4	Endothermic effect in the temperature range of 29–292 °C. Extreme point temperature = 170 °C. The maximum reaction rate temperatures are 87 (water removal), 120, 146, 179 (surfactant, dispersant, defoamer, and coalescing agent decomposition), 248, and 280 °C (pentaerythritol decomposition and beginning of APP decomposition). Endothermic effect in the temperature range of 292–402 °C. Extreme point temperature = 329 °C. The maximum decomposition rate temperatures = 304 and 340 °C (melamine decomposition). Exothermic effect in the temperature range of 402–902 °C. Extreme point temperatures = 560, 759 °C. The maximum reaction rate temperatures are 420, 438, 566, and 611 °C (full phosphorus compound decomposition). Mass loss at 500 °C Δm = 78.2%. Mass loss at 600 °C Δm = 90.4%. Total mass loss at 759 °C = 96.7%.
Comp. 5	Endothermic effect in the temperature range of 20–314 °C. Extreme point temperature = 138 °C. The maximum reaction rate temperatures are 88 °C (water removal) and 227 °C (pentaerythritol decomposition). Endothermic effect in the temperature range of 314–376 °C. Extreme point temperature = 335 °C. The maximum decomposition rate temperatures = 324 and 365 °C (melamine decomposition). Exothermic effect in the temperature range of 376–850 °C. Extreme point temperature = 553 °C. The maximum reaction rate temperatures are 494, 511, 545, 557, and 630 °C (full phosphorus compound decomposition). Mass loss at 510 °C = 75%. Mass loss at 594 °C = 86.6%. Mass loss at 685 °C = 98.8%.
Comp. 6	Endothermic effect in the temperature range of 31–380 °C. Extreme point temperatures = 134 °C and 335 °C. The maximum reaction rate temperatures are 89.9 °C (water removal), 235 °C (pentaerythritol decomposition), 320 °C, 335 °C, and 360 °C (melamine decomposition and beginning of phosphorus compound decomposition). Exothermic effect in the temperature range of 380–690 °C. Extreme point temperature = 477 °C. The maximum reaction rate temperatures are 477 °C, 490 °C, 530 °C, 547 °C, 592 °C, and 622 °C (full APP decomposition). Mass loss at 407 °C = 68.9%. Mass loss at 600 °C = 87.9%. Mass loss at 690 °C = 96.6%.

**Table 3 materials-15-00011-t003:** The main grades and indicators of polyvinyl acetate dispersions available on the Russian market.

Indicator Name	Grade Requirements			
	D51S	DF51/10S	DF51/10SL	DF51/15S
1. Dispersion appearance	White or light yellowish viscous liquid, without clumps or foreign inclusions, with a particle size of 1–3 µm. Surface film acceptable.			
2. Mass fraction of residual monomer (vinyl acetate), %, max.	0.48	0.48	0.40	0.48
3. Mass fraction of dry residue, %, min. (a) non-plasticized (b) plasticized	50	51 53	51 53	51 54
4. Relative viscosity in accordance with the standard high-molecular-weight compound cup, s (a) non-plasticized (b) plasticized	11–20	11–40 11–40	16–25 16–25	11–25 16–40
5. Hydrogen ion concentration indicator (pH)	4.7–6.0	4.5–6.0	5.0–6.0	4.7–6.0
6. Frost resistance in non-plasticized dispersion freeze–thaw cycles, min.	4	4	4	4

**Table 4 materials-15-00011-t004:** Dependence of intumescence ratios on phosphorus compounds from different manufacturers.

Name of Phosphorus Compound Included in the Formulation of Flame Retardant	Intumescence Ratio Indicators
PC Exolit AP 422	39
PC Novoflam APP Alinova	20
PC Novoflam APP	32
PC Sample JLS-APP	32
PC Sample JLS-APP 102	5
PC Sample JLS-APP SPICEAL	4
PC Sample JLS-APP 103	28
PC FR CROS 484	39
PC FR CROS 282	7
PC Exflam APP-201	28
PC PHOS-CHEK P42	21
PC PHOS-CHEK P42C	27
PC PHOS-CHEK P30	30
PC ANTIBLAZE	27
PC Exolit AP 422	39

**Table 5 materials-15-00011-t005:** Contact angles.

Filler Name	Contact Angle	Hydrophilicity/Hydrophobicity
Dicyandiamide	180	Hydrophilic
Melamine	180	Hydrophilic
Chlorinated paraffin	<90	Hydrophobic
Diammonium phosphate	180	Hydrophilic
Urea	180	Hydrophilic
Phosphorus compound (Spain)	180	Hydrophilic
Pentaerythritol	180	Hydrophilic
Sorbitol	180	Hydrophilic
Titanium dioxide	180	Hydrophilic
Dipentaerythritol	180	Hydrophilic
Starch	180	Hydrophilic
Phosphorus compound (Germany)	180	Hydrophilic

**Table 6 materials-15-00011-t006:** Phosphate properties.

Phosphate Name	Water Solubility	Decomposition Temperature, °C	Primary Decomposition Products	Solution (Suspension) pH
NH_4_H_2_PO_4_	Soluble	147	NH_3_, H_3_PO_4_	3–4
(NH_4_)_2_HPO_4_	Soluble	87	NH_3_, H_3_PO_4_	8–9
Urea phosphate	Soluble	130	NH_3_, H_3_PO_4_, CO_2_, H_2_O	9–10
Ammonium polyphosphate	Insoluble	240	NH_3_, (HPO_3_)n	6–7

**Table 7 materials-15-00011-t007:** Foaming agent properties.

Foaming Agent Name	Water Solubility	Decomposition Temperature, °C	Primary Decomposition Products	Solution (Suspension) pH
Urea	Soluble	130	NH_3_, CO_2_, H_2_O	8–9
Thiourea	Poorly soluble	96	NH_3_, CO_2_, H_2_O, SO_2_	6–7
Dicyandiamide	Poorly soluble	180	NH_3_, CO_2_, H_2_O	7–8
Melamine	Insoluble	300	NH_3_, CO_2_, H_2_O	7–8

**Table 8 materials-15-00011-t008:** Carbonizing substance properties.

Carbonizing Substance Name	Water Solubility	Decomposition Temperature, °C	Solution (Suspension) pH
Starch	Soluble	140	7.0
Sorbitol	Soluble	110	6–7
Pentaerythritol	Poorly soluble	263.5	6–7

**Table 9 materials-15-00011-t009:** Halogenated substance properties.

Halogenated Substance Name	Water Solubility	Decomposition Temperature	Solution (Suspension) pH
Chlorinated paraffin	Insoluble	160–350	5–6
Chlorine-containing polymer (KhSPEL)	Insoluble	140	5–6

**Table 10 materials-15-00011-t010:** Main formulations of new flame retardants.

Component Description	Comp. 1	Comp. 2	Comp. 3	Comp. 4	Comp. 5	Comp. 6
Water	19.7	23.8	20.1	20.1	16.5	20.1
Carbonizing agent	6.0	9.6	12.0	12.0	11.1	12.0
Intumescent agent	7.0	5.4	12.0	12.0	12.9	12.0
Phosphorus compound	28.0	21.4	24.0	24.0	22.9	24.0
Polyvinyl acetate dispersion	20.0	20.0	20.0	20.0	22.4	-
Property modifier set	19.3	15.3	11.9	11.9	11.3	11.9

**Table 11 materials-15-00011-t011:** Plan and test results of the 2nd system.

Matrix X								Matrix Y			
x_1_	x_2_	x_3_	x_4_	x_5_	x_6_	x_7_	x_8_	Y_1_	Y_2_	Y_3_	Y_4_
0.5	0.5	0.5	0.5	0.5	0.5	0.5	0.5	10	120	15	1
1.5	1.5	0.5	0.5	1.5	1.5	0.5	0.5	10	48	15	1
1.5	0.5	1.5	0.5	1.5	0.5	1.5	0.5	10	48	15	1
0.5	1.5	1.5	0.5	0.5	1.5	1.5	1.5	10	24	5	2
1.5	0.5	0.5	1.5	0.5	1.5	1.5	0.5	30	24	5	1
0.5	1.5	0.5	1.5	1.5	0.5	1.5	0.5	30	24	3	2
0.5	0.5	1.5	1.5	1.5	1.5	0.5	1.5	10	24	3	2
1.5	1.5	1.5	1.5	0.5	0.5	0.5	0.5	50	48	15	1
1.5	1.5	1.5	1.5	1.5	1.5	1.5	1.5	80	24	3	2
15	30	16	16	7	3.5	5	3	As = 0.948	1.57	0.146	0.187
9	20	10	10	4	0.5	1.5	1.5	Ex = −0.765	2.24	−2.14	−2.17

where **x_ki_** = 0.5 + (x_ki_ − x_kmin_)/(x_kmax_ − x_kmin_), k Є [1.8]; I Є [1.9].

**Table 12 materials-15-00011-t012:** Comparison of the experimentally obtained values.

No.	y_1_^e^	y_1a_^c^	y_1c_^c^	y_1b_^c^	y_1d_^c^	y_2_^e^	y_2a_^c^	y_2b_^c^	y_3_^e^	y_3a_^c^	y_3b_^c^	y_3c_^c^	y_3d_^c^	y_4_^e^	y_4a_^c^	y_4b_^c^	y_4c_^c^	y_4d_^c^
1	10	0.43	10.6	18.9	6.84	1	1.03	0.997	15	15	15.4	14.7	15.1	120	121	120	118	120
2	10	6.2	20.5	10.4	8.22	1	0.959	0.999	15	14.8	15.1	15.8	15.2	48	35.9	42.6	47.5	47.9
3	10	20.7	13	9.03	7.85	1	0.983	0.997	15	14.7	14.3	15	14.7	48	45.6	53	49.9	47.9
4	10	5.13	11.4	9.03	10.2	2	2	1.98	5	5.13	4.06	5	5.04	24	27.5	23.3	26.3	26.4
5	30	28.8	20.3	27.1	31.2	1	1.05	1.03	5	4.97	4.32	5	4.95	24	35.9	31.3	27.1	25.5
6	30	30.6	20.5	27.1	29.8	2	2	1.99	3	0.73	4.06	3.02	1.98	24	7.78	22.5	19.3	23
7	10	3.83	13	9.03	10.5	2	2	2.02	3	3.28	3.28	1.75	2.26	24	27.5	24.9	26.3	22.3
8	60	55.8	59.7	59.2	60.1	1	0.983	1.03	15	14.8	15.1	14.7	15.1	48	45.6	38.5	49.9	47.1
9	80	81.3	81.2	80.7	80.1	2	2	1.98	3	4.94	3.28	2.87	3.25	24	23.8	27.8	19.3	24
F	-	15.1	12.7	46.6	136	-	277.6	377.7	-	27.1	65.7	89.3	97	-	11.6	27.1	77.1	384.5

## Data Availability

The data presented in this study are available on request from the corresponding author.

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
