# Peer review of "Method of Mathematical Modeling for the Experimental Evaluation of Flame-Retardant Materials’ Parameters"

_materials, 2021, doi:10.3390/ma15010011_

Round 1

Reviewer 1 Report

The title is inconsistent with the text of manuscript, and suggest contain the mathematical model. Moreover, the format of manuscript need be changed to satisfy the requirement of journal, and not meet the requirement of English science and technology.

Special comments:

(1) Please amend the Russia to English in the Figure 1.

(2) Please mark the meaning of each curve in Figure 2.

(3) what are the aqueous flame retardants?

(4) How to calculate the intumescence ratio of flame retardant?

(5) Please amend the horizontal coordinate and mark the representative meaning of curve in the Figure 3.

(5) What is the relation between mathematical model and the formulation, TGA properties, contact angle and intumescence ratio?

Author Response

1) Please amend the Russia to English in the Figure 1.

changes were made

(2) Please mark the meaning of each curve in Figure 2.

changes were made

(3) what are the aqueous flame retardants?

it is water-soluble binder

(4) How to calculate the intumescence ratio of flame retardant?

The swelling coefficient K is defined as the ratio of the swollen layer thickness h to the initial coating thickness h0: K = h / h0. Measurement of the layer thickness h0 is carried out in three sections of the sample. Swelling coefficients are determined as the arithmetic mean of three measurements

(5) Please amend the horizontal coordinate and mark the representative meaning of curve in the Figure 3.

changes were made

(5) What is the relation between mathematical model and the formulation, TGA properties, contact angle and intumescence ratio?

The mathematical model is associated with the optimization of the formulation of fire retardants (different quantitative ratio) and the properties of fire retardants (adhesion, durability, fire retardant efficiency).

Input parameters - quantitative ratio (different component content percentages) in the formulation of compositions (input parameters);

Output parameters - properties (output parameters).

One of the components is a pigment, the choice of which is made on the basis of properties (Table 4) - contact angle, TGA, hydrophilicity.

For example, to obtain a good latex covering film, it is necessary to use a hydrophilic pigment that stabilizes the aqueous dispersion. The pigment, in turn, affects the swelling factor. Therefore, an analysis was made of intumescent systems, which are divided into four groups: a) polyols - organic hydroxyl-containing compounds with a high carbon content; b) inorganic acids or substances that release acid at 100-250 ° C; c) organic amines or amides; d) halogenated compounds. The determination of hydrophilicity was reduced to the determination of the contact angle of the main fillers used in intumescent fire protection products. The studies are presented in Table 4.

Reviewer 2 Report

The theme of this paper is interesting and important in this field of science. The style of this manuscript is totally different from that of the general scientific paper. Major revisions are needed for acceptance.

1. Introduction part.

In this part, the background, general information, and the aim or purpose of this work should be cited. In this manuscript, the half part of the experimental section should be moved to introduction part. The aim of this work should be shown clearly at the end of this part.

2. Experimental part.

The experimental procedure should be shown in this part. The properties of chemical substances, experimental procedure, measurements, and analyses should be shown in this part. A totally revision is required.

3. Results part.

The results of experiments should be shown in this part.

Please show the solvent for contact angle measurement. If water was used, the contact angle 180 degree means "hydrophobic" but not "hydrophilic".

4. Discussion part.

The discussion based on the experimental results should be shown in this part. The authors showed some parameters without further expression at the present stage. More precise discussion is needed.

5. References part.

The references should be cited in world-wide.

References should be addressed in the order of first-looking in the text. Namely, the number of reference should be started from #1 at the top of the manuscript.

Figures.

The letters should be English alphabet.

The unit for temperatures should be noted in the normal way.

Especially, the note for line and circles should be addressed in Fig. 3.

Tables.

The tables should be shown compactly and clearly in the sophisticated style.

Author Response

All your comments have been taken into account.

Answer to point 3. We use "hydrophilic" because hydrophilicity is a characteristic of the intensity of the molecular interaction of a substance with water, the ability to absorb water well, as well as a high wettability of surfaces with water. Along with hydrophobicity, it refers both to solids, in which it is a surface property, and to individual molecules, their groups, atoms, ions.

Hydrophilicity is characterized by the magnitude of the adsorption bond of the molecules of a substance with water molecules, the formation of indefinite compounds with them, and the distribution of the amount of water according to the values ​​of the bond energy.

Round 2

Reviewer 1 Report

This revised manuscript can be published.

Author Response

Thanks for your comment and participation in the publication of our article

Reviewer 2 Report

The style of manuscript is improved as a normal research paper.

I still wonder the definition of contact angle and hydrophilicity. What is the solvent for contact angle measurement? Hydrophilicity would be evaluated by the contact angle against water. The readers would be confused why the contact angle of water is too high but it is assigned as hydrophilic.

Author Response

To answer your question, we have made changes to the text of the article. Please see the attachment
